

# AI Image-based method for a robust automatic real-time water level monitoring: A long-term application case

Xabier Blanch[1,2], Jens Grundmann[3], Ralf Hedel[4], Anette Eltner[1]

[1] Institute of Photogrammetry and Remote Sensing, Dresden University of Technology, Dresden, Germany
[2] Department of Civil and Environmental Engineering, Universitat Politècnica de Catalunya-BarcelonaTech, Barcelona, Spain
[3] Institute of Hydrology and Meteorology, Dresden University of Technology, Dresden, Germany
[4] Fraunhofer-Institut für Verkehrs- und Infrastruktursysteme IVI, Dresden

*Correspondence to:* Xabier Blanch (xabier.blanch@upc.edu)

**Abstract:** The study presents a robust, automated camera gauge for long-term river water level monitoring operating in near
real-time. The system employs artificial intelligence (AI) for the image-based segmentation of water bodies and the identification of ground control points (GCPs), combined with photogrammetric techniques, to determine water levels from surveillance camera data acquired every 15 minutes. The method was tested at four locations over a period of more than 2.5 years. During this period over 219,000 images were processed. The results demonstrate a high degree of accuracy, with mean absolute errors ranging from 1.0 to 2.3 cm in comparison to official gauge references. The camera gauge demonstrates
resilience to adverse weather and lighting conditions, achieving an image utilisation rate of above 95% throughout the entire period. The integration of infrared illumination enabled 24/7 monitoring capabilities. Key factors influencing accuracy were identified as camera calibration, GCP stability, and vegetation changes. The low-cost, non-invasive approach advances hydrological monitoring capabilities, particularly for flood detection and mitigation in ungauged or remote areas, enhancing image-based techniques for robust, long-term environmental monitoring with frequent, near real-time updates.

## 1 Introduction

The use of image-based systems has transformed the field of geosciences, offering precise and efficient tools for the monitoring and analysis of environmental phenomena. The integration of cameras and photogrammetry in geoscientific research enables the continuous collection of real-time data, facilitating the study of dynamic processes and the acquisition of detailed information on changes in landscapes and ecosystems. These observation systems have been demonstrated to be particularly
beneficial in the monitoring of rivers (Eltner et al., 2018; Manfreda et al., 2024), rock and glacier landscapes (Blanch et al., 2023a; Ioli et al., 2023), soil surface (Epple et al., 2025) and vegetation evolution (Iglhaut et al., 2019) among others. They offer a robust and less intrusive alternative to traditional methods, and their low cost and straightforward implementation (Blanch et al., 2024) make them suitable for deployment in remote or less developed areas, thereby expanding the scope of monitored elements and reducing vulnerability to natural disasters.


In particular, the utilization of image-based systems for river monitoring offers a number of advantages over the use of conventional gauging stations. These include greater flexibility in camera placement, the ability to monitor multiple points of a river simultaneously, and reduced costs associated with system installation and maintenance. Additionally, cameras allow for data acquisition in adverse conditions and at very short time intervals, providing a comprehensive and uninterrupted
perspective of riverine fluctuations. These advantages make image-based systems an optimal tool for the management and study of water resources, thus increasing the capacity to respond to extreme events and facilitating decision-making in water management.



Monitoring river water levels is a basic but fundamental metric for understanding river behaviour and having this information

in real time is crucial for managing flood risks. The ability to detect, predict and mitigate the consequences of water level changes is immensely useful for disaster managers and for minimizing the impacts to the community. Therefore, the development of low-cost, automated water level detection systems, i.e., camera gauges, is essential, as they provide accurate and continuous data, enabling early warning systems and significantly improving the response to critical events (Manfreda et al., 2024).


On top of that, the integration of artificial intelligence (AI) in hydrological monitoring, while rapidly advancing, has yet to be fully explored for sustained real-time water level detection. The advancement of AI allows the automation of processes related to image processing and information extraction, transforming these image-based systems into truly automatic and intelligent systems capable of providing valuable results in near real time. In the case of water level detection from images, the use of AI

has been a step forward, for example by automatically segmenting water pixels in images, evolving the RGB information in the images into an automatic interpretation of their content, more accurately and faster than traditional computer vision methods (Akiyama et al., 2020).

While camera gauges have shown promise, their long-term reliability and performance under varied environmental conditions

remain challenging. Traditional methods often struggle with continuous operation, particularly during adverse weather conditions or at night.  The literature discusses various automatic methods for detecting water levels from images. For instance, one approach is to install scale bars in the observation area for an automatic measurement based on estimating the contact of the water with the scale bar (Kuo and Tai, 2022; Pan et al., 2018). These methods are highly efficient and provide good accuracy, but they require intervention in the river to install the scale bar, involving logistical challenges and maintenance

issues, which limit the system's versatility and potential for widespread deployment, especially in natural environments.

Another approach involves transforming the scale bars into landmarks, which are points present in the image with known elevations (e.g., obtained through ground surveys). This approach, known as landmark-based water-level estimation (LBWLE by Vandaele et al., 2021), requires identifying these elements in the image and performing interpolation between the two

elevations. The accuracy of the measurements is directly related to the ability to identify known elevation points in the images, and the linear interpolation may not correspond to the actual elevation distribution in the images. Another method that does not require any field installation is the Static Observer Flooding Index (SOFI) method (Moy de Vitry et al., 2019), which detects water level variation based on a direct correlation between the number of pixels segmented as water in each image. This method does not provide direct metric values of the water level but does allow the identification of trends during flooding

events (Vandaele et al., 2021).

Other approaches, like the one developed in this article, use the strategy of image-to-geometry registration, which involves reprojecting automatically segmented images into 3D models containing metric real-world information. This method, extensively discussed by Elias et al. (2019) and Eltner et al. (2018), enables the estimation of water levels as real-world

elevations by establishing a correspondence between 2D image pixels and their corresponding 3D coordinates in a metrically scaled model of the environment.

Works based on this approach include Eltner et al. (2021), who laid the groundwork for this study; Zamboni et al. (2025), who aimed to estimate water levels using image-to-geometry and by leveraging deep learning segmentation models that minimize

the need for annotated datasets, lowering the effort and the computational cost of the image water segmentation; Erfani et al. (2023) who applied the AI and image-to-geometry approach but for a very short time period (sub-daily); and Krüger et al.



(2024) who use a low-cost Raspberry Pi-based camera system a to estimate water levels using the approach developed in this study in a flash flood environment. Additionally, Elias et al. (2019) developed a smartphone application, "Open Water Levels", that utilizes image-to-geometry registration to enable citizen scientists to capture water level measurements using the
smartphone as a measuring device.

However, while the aforementioned studies have shown the potential for hydrological monitoring with cameras, none of them address the operational aspects of the system (i.e., long-term use), as they are limited to specific study areas and short-term observations. Moreover, these approaches still face certain limitations that are well-known in the image-based systems, which
primarily concern robustness and adaptability of these methods under challenging environmental conditions and nighttime observations. The work presented here addresses these challenges by meeting the robustness criteria defined by Peña-Haro et al. (2021), as it achieves key properties such as continuous image capture throughout the whole day, applicability across different rivers, and the capacity for near real-time data transmission and processing.

To address these challenges, new research is relying on AI solutions to bring robustness to image processing. Object segmentation in images using convolutional neural networks (CNNs) has become an essential tool in data analysis, especially in applications requiring detection of features in natural and urban environments. CNNs are a class of deep learning models designed to process grid-like data structures, such as images, leveraging convolutional layers to extract features and patterns from the inputs. In the case of image segmentation these networks classify each pixel in an image, identifying different
elements within the scene (e.g., water bodies in our case). The use of neural networks for image segmentation represents a significant improvement in results, enhancing the performance of traditional computer vision algorithms (Moghimi et al., 2024).

Two recent studies have explored the segmentation of water bodies. Wagner et al. (2023) tested 32 neural networks on the
RIWA dataset – a dataset specifically created to segment water in rivers for monitoring purposes - (Blanch et al., 2023b). Moghimi et al. (2024) evaluated the performance of six modern neural networks across different datasets, including the RIWA dataset. In both works, the U-Net neural network (Ronneberger et al., 2015) was selected as the best-performing model for the RIWA dataset. In Wagner et al. (2023), the UPerNet (Xiao et al., 2018) neural network demonstrated a similar accuracy to U-Net but with a considerable reduction in loss, ensuring higher quality during inference. In line, Wang et al., 2024 have recently
tried the ResUnet + SAM framework to segment water images (including RIWA dataset) in order to monitor the water level trend in UK rivers.

In addition to CNNs for water body segmentation, Zamboni et al. (2025) take advantage of the fixed camera systems to evaluate the use of Space-Time Correspondence Networks (STCN), treating each image as a new frame in a sequence and identifying
the difference, thus avoiding the cost of training a specific water body model. They also compare the results using generic models such as Segment Anything – SAM (Kirillov et al., 2023), showing a notable loss of accuracy with respect to the reference. Although these generic approaches offer convenience and cost savings, their accuracy limitations make them unsuitable for reliable water level monitoring. Therefore, in this work we employ advanced in-house trained AI models to robustly segment water bodies in automatically acquired images.

By combining AI techniques – for image segmentation and ground control point detection (GCP) - with established photogrammetric methods for image-to-geometry registration, our approach enables consistent and accurate water level monitoring over extended observation periods. This study reflects the evolution from initial proof-of-concept testing (Eltner et al., 2018, 2021) to the development of an operational system, capable of obtaining results 24/7, even in adverse weather





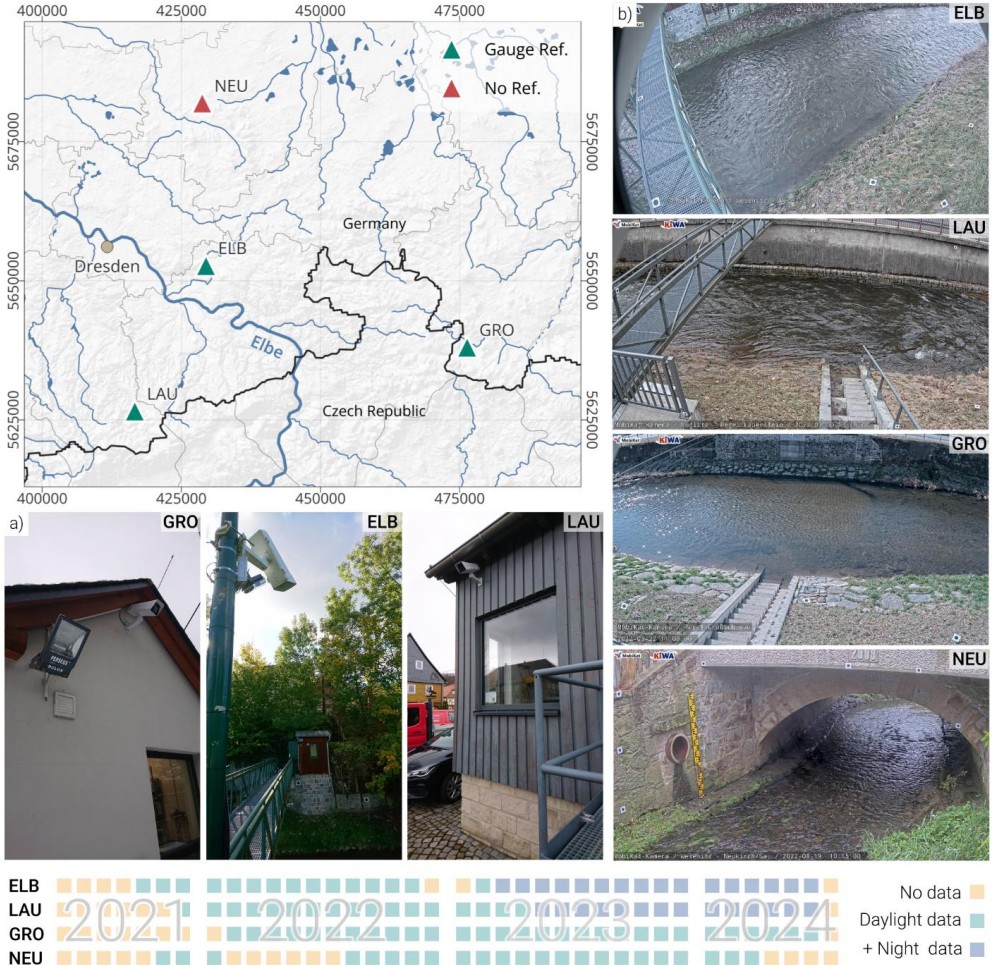

**Figure 1. Map with the distribution of study sites and camera placement. A) Camera installation at GRO, ELB and LAU study sites. B) Images captured by each camera providing a view at the study area. The lower part of the figure shows the periods in which data was captured at each study area.**

conditions. Our work not only demonstrates the viability of image-based systems but also provides a significant advancement in very low-cost hydrological monitoring, offering a robust solution that addresses the key challenges of continuous operation and reliability in real-world conditions.

## 2 Methodology

### 2.1 Study area

This research utilizes images captured at four study sites in Saxony, Germany, within the framework of the KIWA project (Grundmann et al., 2024). At three of these sites, the camera is situated at gauging stations operated by the Staatliche Betriebsgesellschaft für Umwelt und Landwirtschaft, the Saxon state company for the environment and agriculture, enabling comparison of the measurements to reference gauges. These three stations are Elbersdorf (ELB) at the Wesenitz river, Großschönau 2 (GRO) at the Mandau River, and Lauenstein 4 (LAU) at the Müglitz river. The fourth camera is situated in the village of Neukirch (NEU), also at the Wesenitz river. All four cameras are located on tributaries of the Elbe River (Figure 1).





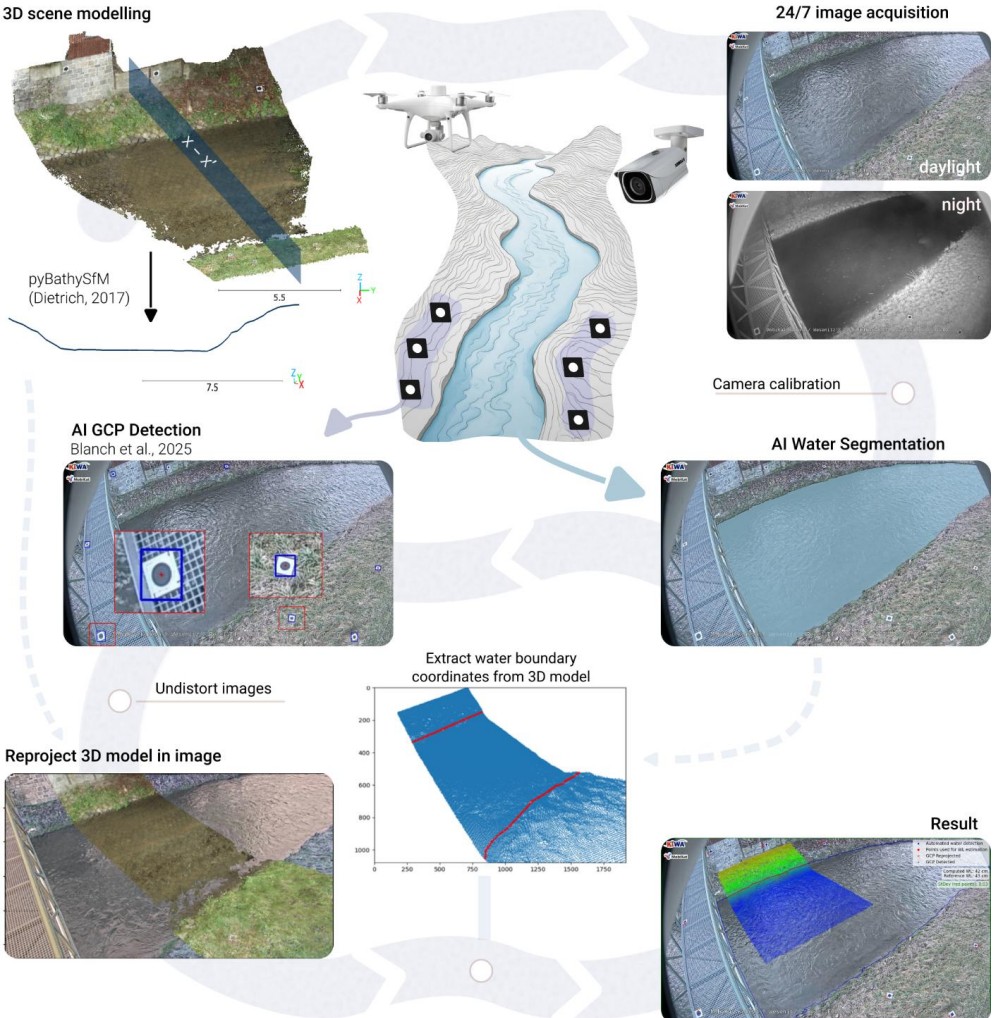

**Figure 2. Graphical workflow for obtaining water level from an AI strategy for segmenting and identifying Ground Control Points (GCPs) and image-to-geometry for obtaining metric values of the water level,**

The images are captured using surveillance cameras, which provide an integrated solution for capturing images and videos and transmitting the data remotely to a server (Figure 1a). At ELB, LAU, and GRO stations, the Axis Q1645 LE camera model is used, while at NEU the Q1615 Mk III camera is employed. Both camera types capture images with a maximum resolution

of 1920x1080, but they differ in pixel size, being 3.75 µm and 2.90 µm, respectively (Table 1). All cameras are equipped with a zoom lens covering focal lengths between approximately 3 to 10 mm, which enables efficient coverage of the study area (Figure 1b).

At all four locations, the cameras are configured to capture a still image every 15 minutes, which is transmitted directly to

servers. Capturing is continuous throughout the day, resulting in 96 images acquired per day. During this research, a remotely controlled infrared (IR) light was installed at ELB and LAU stations to illuminate the study area, allowing the cameras to capture usable images at night and facilitating 24-hour observation cycles. In contrast, at GRO and NEU stations, night-time images are captured but are not usable for water level monitoring. The cameras were installed at the end of 2021 and have



covered more than 2.5 years of data collection. Table 1 summarizes the characteristics of each location, including installation
date and images acquired up to June 30, 2024.

**Table 1. Image acquisition properties for each study area.**

|  | Reference | | Camera | | | IR support | | Number of images | | |
|---|---|---|---|---|---|---|---|---|---|---|
|  | River | HWIMS Ref. | Model | 1st acquisition | Calibration | IR-Lamp | Installation | Day | IR Night | Total |
| **ELB** | Wesenitz | Yes | Q1645 LE | 07 Nov 21 | 3D Model | Yes | 04 Nov 22 | 48,759 | 24,063 | 72,822 |
| **LAU** | Müglitz | Yes | Q1645 LE | 16 Dec 21 | In field | Yes | 03 May 23 | 47,893 | 17,684 | 65,577 |
| **GRO** | Mandau | Yes | Q1645 LE | 17 Feb 22 | In field | No | - | 47,109 | - | 47,109 |
| **NEU** | Wesenitz | No | Q1615 Mk III | 24 Nov 21 | 3D Model | No | - | 33,382 | - | 33,382 |

### 2.2 3D Modelling and Camera Calibration

The first step in the workflow (Figure 2) involves creating a high-resolution 3D model of the study area, covering the river
region monitored by the surveillance cameras. GCPs must be installed within the study area to support photogrammetric data
processing and to retrieve the position and orientation, i.e., exterior orientation parameters, of the camera. The GCPs must be
visible in both the 3D model and the still images captured by the gauge cameras. Furthermore, proper distribution of the GCPs
in the image (covering the outer frame of the image) ensures an accurate estimation of the camera geometry. In this work, the
GCPs were measured with centimetre accuracy using RTK-GNSS. The 3D models were generated using terrestrial
photogrammetry along with UAV photogrammetry when possible, using the Structure-from-Motion Multi-View Stereo (SfM-
MVS) algorithm (e.g., Eltner and Sofia, 2020; Smith et al., 2016; Westoby et al., 2012), and the resulting model was
georeferenced with the fixed GCPs as well as additionally placed temporary GCPs that were also measured with RTK-GNSS

Given that the accurate 3D mapping of the river reach, including the riverbed, is crucial in water level measurement, the
PyBathySfM tool (Dietrich, 2017) was employed to apply refraction correction to areas underwater, which is typically not
considered during standard SfM 3D reconstruction. After correction, a reference Z coordinate in the riverbed of the 3D model
is selected, serving as the zero height for the water surface (h_kiwa0). When 3D reconstruction of the riverbed via
photogrammetry is not possible - due to factors such as high-water levels or water opacity - we capture cross-sections using
RTK-GNSS. In these cases, the underwater area is reconstructed by interpolating these points, resulting in a mesh that directly
represents the elevation (h_kiwa0). This riverbed mesh is subsequently integrated with the SfM model, which captures the
entire surrounding study area

With both approaches, the h_kiwa0 reference is obtained by averaging the Z coordinates of a square region (e.g. ELB: area
4.5m$^2$) located in the central part of the river bed, and it may differ from the reference zero height (h_ref0) of the official
gauging stations. To resolve this mismatch, an offset is incorporated into the calculated values (wl_kiwa). This offset is the
one that minimises the differences with respect to the reference values (wl_ref) across the entire time series, and for the period
presented in this study, it has values of ELB: -2.0 cm, LAU: -1.0 cm, and GRO: -2.5 cm.

Fieldwork includes camera calibration, i.e., the estimation of the interior camera parameters focal distances, principal point
and distortion parameters. Before installing the cameras on their final mounts, images of a calibration chart were captured from
different perspectives to avoid parameter correlation during the camera calibration (e.g., Liebold et al., 2023). The interior



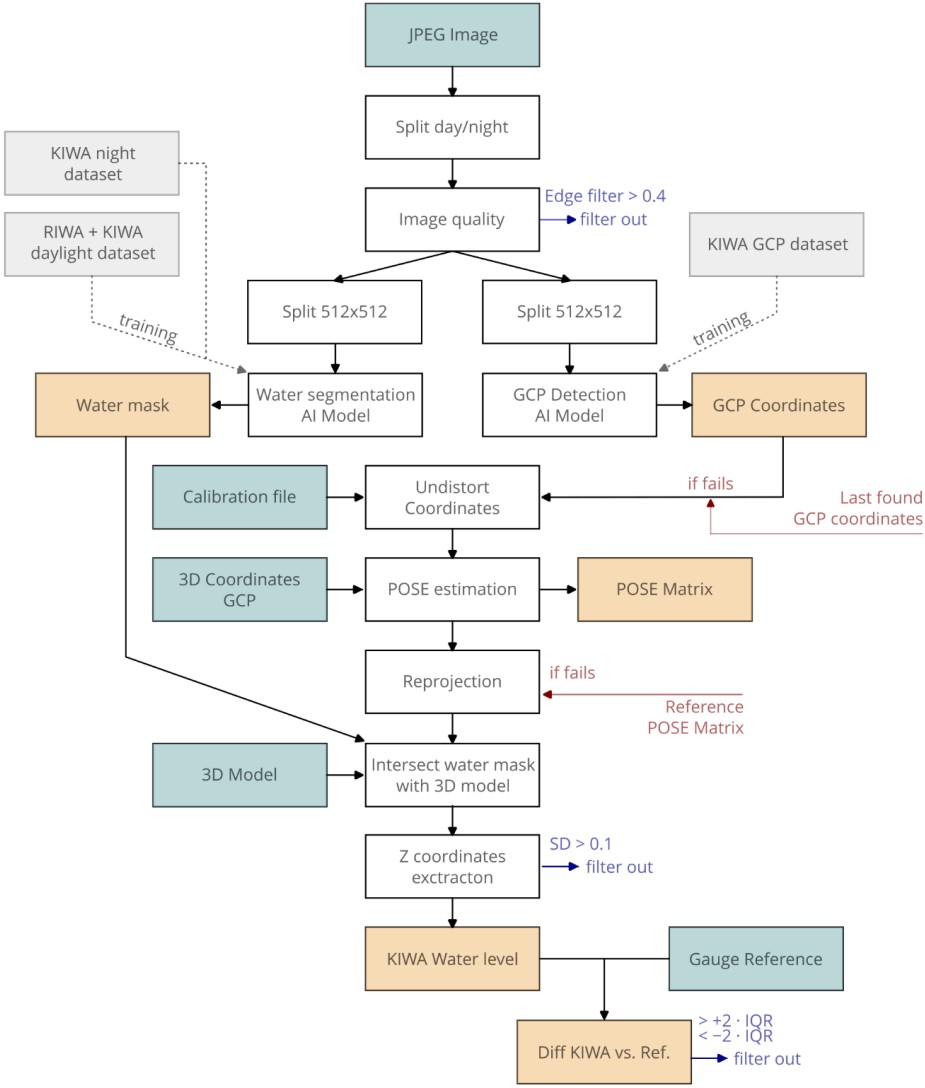

**Figure 3. Flowchart summarizing the complete process from image acquisition to water level determination. The chart outlines the steps from the initial JPG image captured by the camera to the final extraction of water level data.**

camera parameter estimation was performed using Agisoft Metashape (v2.0.1), incorporating the images of the calibration board and the 3D coordinates of the coded targets on the board in a bundle adjustment. In cases where calibration images could not be captured (i.e., if the camera was already mounted), an approximate calibration was performed. This involve incorporating still images from the fixed camera into the bundle adjustment used to create the 3D model of the study area. By identifying homologous points between the images used for the 3D modelling and the fixed camera images, we could estimate the interior camera parameters for those images. This approach provided a gross calibration file. At stations LAU and GRO, the calibration could be performed using the calibration chart, while at ELB and NEU the rough calibration strategy had to be used.



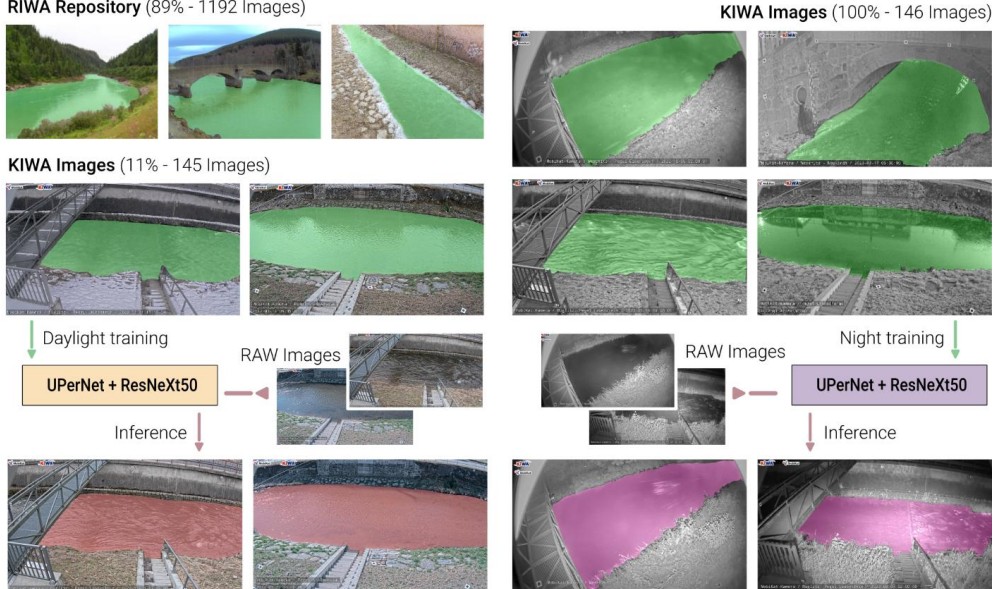

**Figure 4. Data used for AI-based segmentation. Green areas represent manually segmented images, while orange/pink areas show the results obtained from the model for both daytime and nighttime images.**

### 2.3 Image classification and filtering

Surveillance cameras automatically switch to night mode, producing black and white images instead of the colour images used during the day. This change is based on light conditions rather than a fixed schedule. Therefore, the first filtering step involves analysing the RGB channel values in three different pixels of the image (Figure 3). If the values of the three channels are equal in all three selected pixels, the image is classified as nocturnal. This filter allows us to exclude all night-captured images from the daytime processing batch. Night images, in the absence of infra-red (IR) light, are not used because the river's visibility is incomplete and depends on the ambient light conditions of each scenario. If IR lamps are present, night images are processed with the parameters defined for the night images. The next filtering step, applied only to daytime images, involves assessing the sharpness of the images. A Sobel filter ($> 0.4$) is used to determine the average edge detection value of each image. This allows to exclude any image from processing that is blurry, out of focus, or where the region of interest (ROI) is not visible due to adverse weather conditions (e.g., heavy snowfall, fog) (Figure 3).

### 2.4 AI Segmentation

Once the image passes the preliminary filters, we use a CNN to perform image segmentation, i.e., to select the water surface, defined by water pixels, and extract it from the background (Figure 2). For this purpose, we tested 32 different CNN architectures (Wagner et al., 2023) on the River Water Segmentation Dataset, also known as RIWA (Blanch et al., 2023). This dataset comprises 1163 daylight images of rivers captured with smartphones, drones, and DSLR cameras, manually labelled, as well as river images from the WaterNet dataset (Liang et al., 2020) and AED20K images (Zhou et al., 2019). Part of the images used to create the RIWA dataset were also obtained with the KIWA project cameras (22 images, 1.9% of the dataset). We found that the CNN architecture with the best performance for segmenting KIWA images was UPerNet neural network with the ResNeXt50 (Xiao et al., 2018) backbone for the feature extraction.

To improve the training process, the original RIWA dataset is iteratively modified to exclude water images that are significantly different from our target scenario and to include more images captured by KIWA cameras (Figure 4). To determine which



KIWA images to include, an iterative process (Deep Active Learning) is used to select images that are poorly segmented in

previous training runs (Li et al., 2024). Thus, we include challenging or sensitive images that have not been correctly segmented before (e.g., bad weather, transparent water, strong shadows over the water) in subsequent trainings. The final dataset used for the last training consists of 1,337 images, of which 145 (11% of the dataset) are KIWA images covering various locations, weather conditions, and water levels (Figure 4). Similarly, a training dataset is generated for images captured with the infrared sensor (Figure 4). The absence of publicly available infrared image datasets limits the creation of a more general and

transferable dataset, leading to a single dataset of 146 KIWA images, which include various weather conditions and water levels.

Both datasets are augmented using the Albumentation library (Buslaev et al., 2020), which allows for modifying both the original image and the corresponding mask. Data augmentation enables the synthetic creation of larger datasets, resulting in

more robust training and more transferable results across different locations and conditions. Augmentation, especially based on geometric modifications, is crucial when working with fixed cameras as it helps to prevent overfitting due to spatial correlations. Additionally, pixel-level modifications are applied within a realistic range that ensures no unrealistic colours or switches between RGB channels have been introduced, maintaining the natural appearance of the images from our sites. Data augmentation was applied after generating all 512×512 patches from each original image in the training dataset, with each

original patch producing four augmented versions.

For training the UPerNet network with a ResNeXt50 backbone in daytime images, we use the FocalLoss cost function and the Adam optimizer with an initial learning rate of 0.0001, which is later reduced using the ReduceLROnPlateau algorithm. Training is conducted on a NVIDIA GTX A6000, with a batch size of 30 for 1000 epochs. To assess the training metrics, an

independent evaluation dataset of 30 images is used, 21 of which are KIWA images (70%). This imbalance in the evaluation dataset is justified because we aim to optimize the model specifically for KIWA images rather than a random distribution of available images. The best model obtained for daytime images has a segmentation accuracy of 98.9% of correctly identified pixels, while the model obtained for night images has an accuracy of 99.1%. Accuracy represents the proportion of pixels correctly classified as river or riverbank relative to the total number of pixels in the image. Once the models are trained, we

automatically perform image inference (Figure 4), so that every 15 minutes we automatically receive not only the camera image, but also the mask that identifies the areas corresponding to the river and to the riverbanks.

### 2.5 AI GCP Identification

Although the cameras are installed using fixed mounts, stability of the cameras and sensors during the observation period is not given. We detect abrupt and smooth camera movements. These movements, linked to physical disruptions and thermal

disturbances, result in variations of the 2D image coordinates of the GCPs. Identifying these coordinates with high accuracy and reliability is important in the performance of the camera gauge (Figure 2). Therefore, the GCPs need to be measured in each image every time.

Automatic identification of GCPs in images is addressed differently in the literature, for example, using tracking algorithms

(e.g., Eltner et al., 2017), feature descriptors like SIFT or SURF (Chureesampant and Susaki, 2014) or geometric shape-fitting (e.g., Maalek and Lichti, 2021). For this work, we utilize an artificial intelligence-based approach that allows us to directly obtain the centre of each GCP. We adapt and retrain the R-CNN Keypoint detector neural network for the automatic identification of each GCP's coordinates. This method, extensively explained in Blanch et al. (2025) allows to obtain GCP identification with a precision of less than 0.5 pixel, automatically and without any pre- or post-processing of the images or

results. With the model specifically trained on KIWA images, we develop a robust approach to identify GCPs with good

transferability and capable of detecting GCPs in scenarios where the other above mentioned detection algorithms fail (Blanch et al., 2025a).

If the automatic GCP detection system fails to identify up to four GCPs in the image (e.g., due to occlusion or disappearance), the KNNImputer algorithm (scikit-learn) is used to provide an estimated coordinate for the missing GCPs. This algorithm aims to determine undetected values by searching for nearby images (i.e., images where the GCP coordinates are very similar to the current image's GCP coordinates) and assigning the missing values. If our AI-based GCP detector fails to determine more than four GCPs, the coordinates from the previous image where all GCPs were correctly identified are automatically assigned.

**2.5 Photogrammetric process**

Once the 2D coordinates of the GCPs are obtained, along with the calibration file (interior camera orientation parameters), we proceed with the reprojection of the 3D point cloud into the 2D image space (see Elias et al., 2019; Eltner et al., 2018 for more explanation) (Figure 2 and 3). The first step is to compute the parameters of the exterior orientation by determining the camera's location and orientation by correlating the 2D coordinates of the GCPs with their 3D coordinates in the real world (world coordinate system). This is achieved by considering the collinearity constraints to eventually transform points from the real-
world 3D space to the 2D space of the camera. Thereby, the 3D point cloud of the study area is reprojected onto the image plane, requiring the knowledge about the interior camera parameters besides the exterior geometry. Consequently, each pixel in the image that got hit by a reprojected 3D point can be assigned a corresponding 3D coordinate from the world coordinate system. The final step involves extracting the contour lines of the water mask that delimits the pixels classified as water by the AI segmentation approach and using the nearest neighbour algorithm to find which points in the reprojected 3D point cloud
are closest to this line. Only the upper contour of the segmented water mask, i.e., the river side opposite to the camera, is used in this study, and the search for nearby points is limited to a central, cropped area of the point cloud to ensure the usage of a reliable part of the 3D model of the river reach.

With the selected points of the 3D point cloud closest to the water line a statistical calculation of the Z coordinates is performed, resulting in a median Z coordinate that determines the water surface position (h_kiwa). Finally, by subtracting the reference elevation and adding the right offset (h_kiwa – h_kiwa0 + offset), the water level (wl_kiwa) is obtained. Additionally, the standard deviation of the distribution of Z coordinate values along the intersected points is used as a parameter to estimate the quality of the water level measurement.

At the locations with a gauging station (ELB, LAU, and GRO), the wl_kiwa values is compared to the values made available by the official monitoring network (wl_ref). These reference values are obtained using float operated or bubble gauges (redundant measurement system). The reference is a 15-minute average value obtained by averaging measurements in 5-minute interval. Since these values, unlike wl_kiwa, do not correspond to the river's current state but to an average of its behaviour every 15 minutes, the comparisons to determine the method's accuracy are presented on a daily basis, averaging wl_kiwa and
wl_ref every 24 hours.

Filtering of results and outliers is done on top of the obtained results because the workflow is automatic and was directly applied to all images received on the server. Results are filtered based on two statistical criteria. The first is based on the standard deviation value of the Z coordinate of the points intersecting with the 3D model. A high dispersion of this value
indicates that the segmentation boundary is cutting through various Z coordinate levels and hence is incorrect, as we assume horizontal water behaviour. The second statistical filter criterion is applied to the wl_kiwa results to eliminate potential outliers





compared to the wl_ref. We use the modified Tukey filter to identify extreme outliers, utilising two times the interquartile range (Q3-Q1).

## 3 Results

The water level detection system processed data ranging from 660 days (22 months) of the NEU camera to 946 days of the ELB camera (31.5 months). In total, 219,720 images have been automatically processed. Except for the ELB location, images have been mostly obtained continuously throughout the observation period. Generally, the utilization rate (images acquired vs valid water level measurements) ranges from 86.87% for NEU, where the system captures not usable images at night with the daylight sensor, to 99.2% for LAU IR, which is similar to the values at ELB and GRO. Analysing the image processing stage

by stage revealed that the Sobel filter is crucial for excluding images that are typically corrupted or unable to undergo segmentation. This is usually due to poor image quality caused by adverse weather conditions (e.g., fog, heavy snow) or low light levels (e.g., sunset or sunrise). High ambient light in urban environments (at NEU and GRO) can cause the surveillance camera to activate day mode even at night resulting in failed image processing that justify the higher percentage of images deleted in this step. The image quality-based filter ensured a workflow success rate (valid images vs water level measurements)

of over 99% across all locations (Table 2).

**Table 2. Showing the images passing the filtering and calculation processes. From the images acquired by the camera to the images used to calculate a water level. (Night images without IR light are not included in acquired images). The success rate is calculated for all images acquired (even in bad weather) and for images that pass the Sobel filter (which are considered valid images). Note**
**that no Tukey Filter values are given for NEU due to missing reference at this site.**

|  | Acquired Images |  | Sobel Filter |  | Workflow completed |  | St. Dev. Filter |  | Tukey Filter |  | % of success |  |
|---|---|---|---|---|---|---|---|---|---|---|---|---|
|  | Days | Images | Images | % | Images | % | Images | % | Images | % | Acquired | Usable |
| **ELB** | 946 | 48,759 | 48,517 | 99,5 | 48,272 | 99,5 | 48,018 | 99,5 | 47,852 | 99,7 | 98,1 | 98,6 |
| **ELB IR** | 585 | 24,063 | - | - | 24,057 | 99,9 | 23,520 | 97,8 | 23,471 | 97,8 | 97,5 | 97,5 |
| **LAU** | 927 | 47,947 | 47,893 | 99,8 | 47,869 | 99,9 | 47,857 | 99,9 | 47,318 | 98,8 | 98,8 | 98,8 |
| **LAU IR** | 424 | 17,684 | - | - | 17,665 | 99,9 | 17,614 | 99,7 | 17,538 | 99,5 | 99,2 | 99,2 |
| **GRO** | 864 | 47,109 | 46,526 | 98,7 | 46,510 | 99,9 | 46,118 | 99,1 | 44,903 | 97,3 | 95,3 | 96,5 |
| **NEU** | 660 | 33,382 | 30,147 | 90,3 | 30,142 | 99,9 | 29,001 | 96,2 | - | - | 86,9 | 96,2 |

The obtained water level values from the workflow were filtered based on the standard deviation of the intersected Z coordinates (std < 0.1 m) with the percentage of results passing this filter ranging from 96.2% at NEU to 99.9% at LAU. Files filtered out at this stage are mainly due to faulty water segmentation in the image or very low water levels leading to

intersections of the water line with parts of the model that were not well reconstructed (e.g., contact between the riverbed and slopes). Both cases result in irregular intersections with the 3D model, which implies different Z-coordinates and higher dispersion of these values. The final outlier filter (2·IQR; Tukey filter) allowed more than 98.5% of the images to pass through. Filtered images at that stage generally correspond to issues in the photogrammetric process (e.g., errors in determining the GCPs in the images) resulting in water level measurements significantly different from the reference values (Table 2).


Table 3 shows the reprojection errors obtained at each GCP at each study site. The pixel deviations, which are directly related to the camera calibration and photogrammetric processing, allow to infer the quality of the fit of the 3D model to the 2D image. Note that the GCPs were positioned peripherally in the edges of the images where lens distortions are most pronounced and thus more challenging to correct.




Table 4 shows the comparison between wl_kiwa and wl_ref. The Mean-Absolute Error (MAE) value represents the mean of all absolute differences ($\frac{1}{n}\sum |wl\_kiwa - wl\_ref|$) and together with the Root-Mean-Square Error (RMSE) is used to estimate the deviation of our measurement from the reference. The results are provided in two ways: a) image-by-image (i.e., the wl_kiwa measurement from each image is compared with the wl_ref value for the same timestamp) and b) a 24-hour average

in which all available images for each dataset (could be day, night or all day) are averaged daily to a single value (i.e., wl_kiwa and wl_ref values are averaged using all night images, and then compared). The comparison is done by averaging the reference values for the same timestamps used for the KIWA average. The MAE values, measured for the whole period, range from 1.3 cm to 2.7 cm in LAU and ELB, respectively, for an image-to-image comparison, and from 1.0 cm to 2.3 cm in LAU and GRO, respectively, in the daily average values. For all study areas, the Spearman's coefficient ranges between approximately 0.95

and 1 indicating a high level of correlation between wl_ref and wl_kiwa.

**Table 3. Focal distance of each camera and reprojection error, measured in pixels, at each GCP at the different study site.**

|        | Focal Length | GCP 1   | GCP 2   | GCP 3   | GCP 4   | GCP 5   | GCP 6   | GCP 7   | GCP 8   | Average |
|--------|--------------|---------|---------|---------|---------|---------|---------|---------|---------|---------|
| **ELB** | 4,1 mm      | 11.3 px | 17.6 px | 20.6 px | 20.8 px | 16.1 px | 20.0 px | 15.3 px | 18.6 px | 17.5 px |
| **LAU** | 5.7 mm      | 8.7 px  | 2.5 px  | 6.4 px  | 8.6 px  | 8.6 px  | 7.4 px  | 5.0 px  | 3.5 px  | 6.3 px  |
| **GRO** | 4.6 mm      | 19.0 px | 15.4 px | 39.7 px | 30.1 px | 36.0 px | 28.6 px | 33.7 px | -       | 29.0 px |
| **NEU** | 6,6 mm      | 6.2 px  | 6.3 px  | 5.2 px  | 5.3 px  | 2.8 px  | 4.4 px  | 4.4 px  | -       | 4.9 px  |

To check the evolution of the system over time, we calculate the same values for the first 365 days after installing the systems

(Table 5). The results are less dispersed compared to the entire time series, with a MAE of about 1.5 cm for all locations in both the image-by-image analysis and the daily averages.

**Table 4. Results obtained for the entire time series showing the comparison image by image and the daily average of the official values wl_ref and the calculated values wl_kiwa.**

|               | All-time series | | | | | | | |
|---------------|-----------------|--------|---------|--------|--------|---------|---------|--------|
|               | Image-by-image  | | | | 24h average | | | |
| **Location**  | MAE    | RMSE   | St. Dev | ρ      | MAE    | RMSE    | St. Dev | ρ      |
| ELB DAY       | 2.4 cm | 3.2 cm | 3.2 cm  | 0.93   | 2.3 cm | 3.1 cm  | 3.0 cm  | 0.95   |
| ELB IR        | 3.3 cm | 4.1 cm | 3.9 cm  | 0.95   | 3.1 cm | 3.9 cm  | 3.7 cm  | 0.93   |
| **ELB TOTAL** | **2,7 cm** | **3,4 cm** | **3,4 cm** | **0.94** | **2,3 cm** | **3,0 cm** | **3,0 cm** | **0.95** |
| LAU DAY       | 1.3 cm | 1.7 cm | 1.7 cm  | 0.97   | 1.0 cm | 1.3 cm  | 1.3 cm  | 0.99   |
| LAU IR        | 1.3 cm | 1.6 cm | 1.4 cm  | 0,98   | 1,2 cm | 1,5 cm  | 1,3 cm  | 0,97   |
| **LAU TOTAL** | **1.3 cm** | **1.7 cm** | **1.6 cm** | **0.97** | **1.0 cm** | **1.3 cm** | **1.3 cm** | **0.99** |
| **GRO**       | **1,9 cm** | **2,3 cm** | **2,3 cm** | **0.97** | **1,7 cm** | **2,2 cm** | **2,1 cm** | **0.98** |


**Table 5. Results obtained for the first 365 days of each time series showing the comparison image by image and the daily average of the official values wl_ref and the calculated values wl_kiwa. All values correspond to daylight conditions**

|              | First Year (365 days) | | | | | |
|--------------|-----------------------|--------|---------|--------|--------|---------|
|              | Image-by-image        | | | 24h average | | |
| **Location** | MAE    | RMSE   | St. Dev | MAE    | RMSE   | St. Dev |
| **ELB**      | 1,6 cm | 2,0 cm | 2,0 cm  | 1.4 cm | 1.9 cm | 1.9 cm  |
| **LAU**      | 1,7 cm | 2,0 cm | 1,2 cm  | 1,6 cm | 1,8 cm | 0,8 cm  |
| **GRO**      | 1.6 cm | 2,0 cm | 1,9 cm  | 1,5 cm | 1,7 cm | 1,7 cm  |

Figure 5 presents the images generated during the processing at the time of the maximum water level at each location. The

figure includes the original image captured by the cameras, the re-projection of the 3D model used to calculate the Z coordinate,



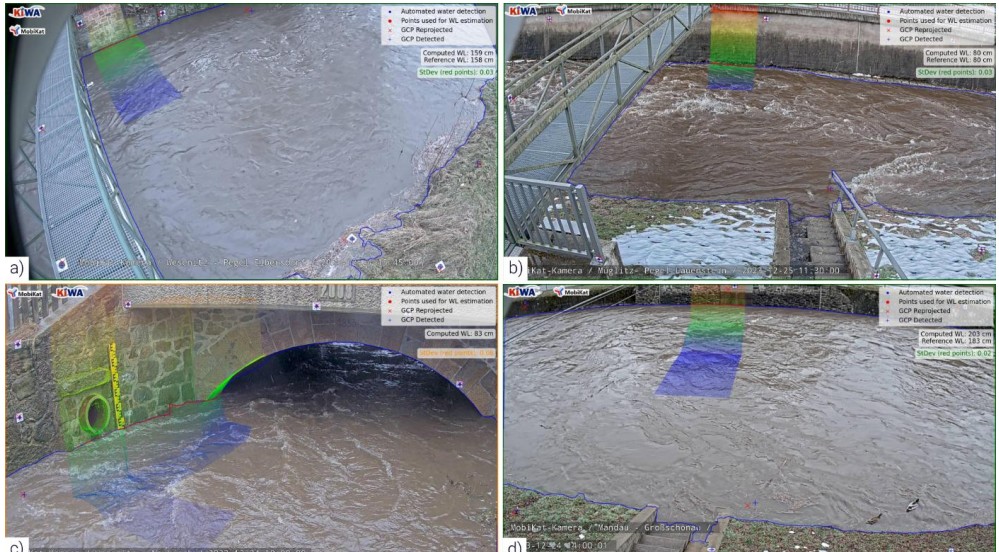

**Figure 5. Frames of each study area at the time of maximum water level. Each image includes the reprojection of the 3D model that is used to calculate the intersection with the water line. In red, the points used for averaging the Z coordinate.**

the red line intersecting the model (indicating where the Z coordinate is measured), the identification and re-projection of the detected GCPs in the image, and, where available (ELB, LAU, GRO), the comparison with wl_ref. A video animation illustrating the water level measurements obtained at the Lauenstein gauging station between December 24 and 31, 2023, is available for viewing in Blanch et al., 2025b (link available in the Video Supplement section).


Figure 6 illustrates the daily comparison between wl_kiwa and wl_ref for the whole time series. For each location, a subplot at the top shows the differences in centimetres for the daily average. The use of colour indicates whether the reference value is being underestimated or overestimated compared to the averaged wl_ref. The difference plot shows that the error distribution between the wl_kiwa and wl_ref remains randomly stable throughout the observation period, with no clear trend of increase

or decrease over time in all three locations. The range of difference along the time series is consistent with the results presented in Table 4, with LAU being the location with the highest accuracy.

The Figure 6 shows identifiable clusters (i.e. periods where the errors are of similar magnitude and direction). These clusters can be attributed to various environmental factors affecting the measurements; for example, during the summer months,

vegetation grows along the riverbanks and occasionally obstructs the monitoring stations until maintenance crews cut the grass. Another pattern is that, although the differences do not tend to increase with rising water levels, the most significant deviations from the reference median are observed during the January 2024 level rises. However, these deviations remain within reasonable limits: 4 cm at LAU for a water level of 70 cm, 8 cm at ELB for a water level above 120 cm, and a difference of 7 cm for water levels above 100 cm. These deviations may be attributed to differences in the methodologies used for water level

measurements, as the reference gauges provide averaged water levels over a period, potentially smoothing out rapid fluctuations typically associated with flood events.

## 4 Discussion

The main contribution of this work is the demonstration of a robust solution for water level measurement using low-cost and non-invasive methods. The systems were tested over an extended period at various study sites. The addition of IR lamps has





**Figure 6. wl_kiwa results, averaged daily, obtained at ELB, LAU and GRO and compared with the value of wl_ref. Blue colours indicate that the reference value has been overestimated, orange being underestimated.**

addressed one of the primary limitations of visual methods, enabling reliable measurements at night and eventually allowing for continuous 24-hour monitoring. By integrating AI into the photogrammetric workflow, the system proves to be highly robust, functioning effectively also under adverse weather conditions (Figure 7). The utilization rate of images is very high (average of 98%), indicating that the system rarely fails to resolve a valid water level. Failures typically occur during periods of extremely poor visibility (e.g., fog, heavy snowfall), when the image-based method is not viable due to inadequate

observation of the water surface (Figure 8ab).



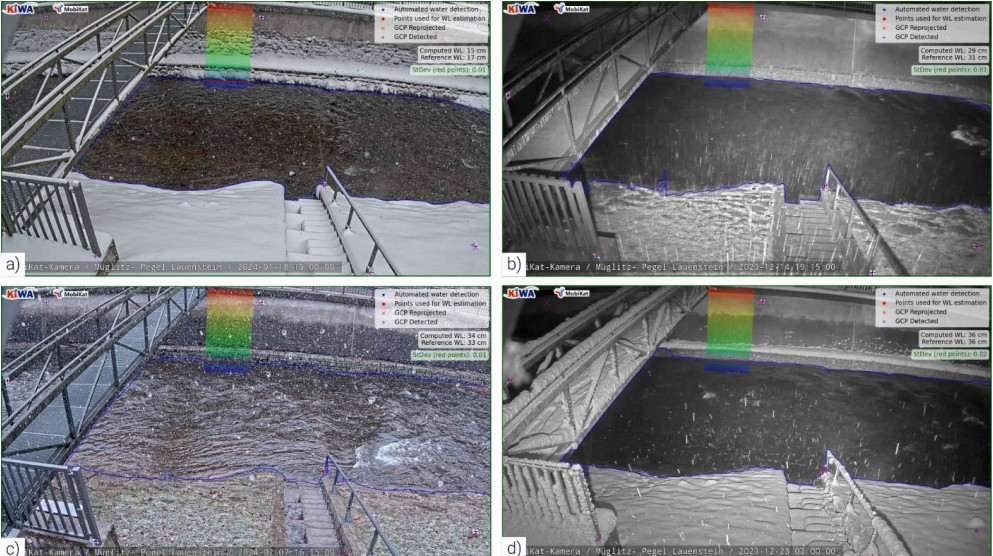

**Figure 7. LAU images obtained during adverse weather conditions correctly resolved by the workflow. a) daytime image captured during snowfall. b) nighttime image captured during rainfall. c) daytime image captured during rainfall. d) nighttime image captured during snowfall. In all cases, the calculated WL value is considered valid and passes the established filters when is compared against the official reference.**

The success in obtaining water level values under adverse weather conditions is due to our iterative training process of the neural network for water segmentation in images. We developed a model that can handle a wide variety of situations effectively including images that the AI initially struggled to resolve in successive training sessions. It should be noted that KIWA images
constitute less than 11% of the training dataset, suggesting the model's high transferability. For future installations with different cameras and environments, we expect that only a small number of manually labelled images will be needed to adapt the model to the new site.

The segmentation of images for precise water level measurement presents a significant challenge (i.e., Moghimi et al., 2024;
Wagner et al., 2023) because it requires precise identification of the interface between the water body and the riverbanks in the image. This boundary is especially difficult to detect accurately in automatic segmentation processes, since on the one hand it is a natural boundary (i.e., waves, water transparency, vegetation), and on the other hand standard metrics such as the Intersection over the Union (IoU) or the Dice coefficient, and therefore the most common models, give priority to the precise segmentation of the whole object rather than to the precise definition of the contour. In a river context, this delineation is
especially challenging due to water transparency at the boundaries, which makes clear delimitation difficult even for human observers. Although we did not employ a contour-centred metric such as the Boundary F1 Score (BFScore), testing various neural network models (Wagner et al., 2023) has allowed us to identify the optimum for correct water segmentation, and thus obtain water level measurements.

Training with night time images proves to be more feasible and accurate because the contrast between the water and the background is more pronounced. Additionally, there are no transparency issues due to the high absorption of infrared light by water, which simplifies the segmentation process. The precision obtained during training for the best-performing model (around 99%) is consistent with the model's performance throughout the entire time series, demonstrating its ability to consistently segment the water bodies in most images. Ensuring that the mask intersecting with the 3D model accurately



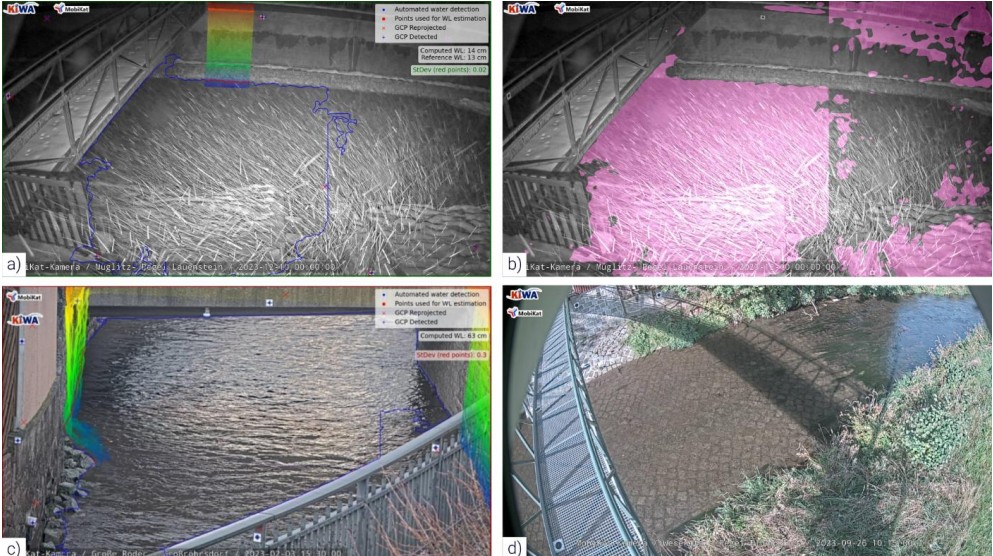

**Figure 8. Main limitations of our camera gauge. a) and b) images from LAU captured during intense snowfall. Although the water level result is correct, the overall AI segmentation is erroneous. c) Image from a study site not presented in this work, where camera calibration was not well resolved, thus preventing the application of the image-to-geometry algorithm. d) Image from ELB showing vegetation occupying both riverbanks, which affects the delineation of the water body and obscures most of the GCPs.**

represents the water boundary at the moment of image acquisition. The results obtained are in line with the ones provided by Wagner et al., (2023) and Moghimi et al., (2024) in their respective research.

The automatic identification of GCPs in each image prevents the accuracy from deteriorating over the observation period as effects by experienced camera movements, e.g. due to thermal effects on the camera (Elias et al., 2020), are mitigated. The

main issue with the use of GCPs has been their durability in the study areas. Although most remained throughout the 2.5-year study, some were damaged or displaced during high-flow periods or shaded temporally due to vegetation cover, indicating that even with an automated camera gauge, maintenance is necessary. Regarding this maintenance, the difference between table 4 (whole time series) and table 5 (one year) shows how the error between the KIWA values and the reference is higher the longer the time window, and although this difference is multifactorial, undoubtedly the deterioration of the GCPs as well as the age

of the camera calibration play an important role. Also, maintenance is necessary due to other key factors such as vegetation. Our results indicate that vegetation significantly impacts deviations to the references by either obscuring GCPs or covering the water-shore contact, leading to irregular segmentation. An effective solution involves automatic regeneration of the 3D models updating the model to the real riverbank situation; however, a crucial aspect of ensuring the robustness of the method is to use measurement areas where the water-slope contact remains unobstructed over time. Figure 6 illustrates how, particularly during

summer, sudden changes lead to the system underestimating or overestimating values (notably in ELB, July 2023). These anomalies are related to vegetation changes, with abrupt shifts corresponding to days when landscaping work was performed. Conversely, stations like LAU show more stable values and greater accuracy throughout the year, as the water-shore contact occurs on a concrete wall, providing greater stability over time.

In addition to the accurate segmentation of water and the identification of GCPs, another crucial element for ensuring good water level measurement is the proper calibration of the fixed cameras (Figure 8c). This was challenging if the cameras were already installed and operational. In our case, we observe that the ELB camera, which was already in operation and calibrated approximately (i.e., adding the image into the SfM bundle adjustment), shows slightly different precision compared to LAU,



where a much more precise calibration was performed (i.e., field board calibration). GRO also underwent field calibration, but

the use of a very short focal length and the large distance of the ROI to the camera adversely affects precision. During the two

years, the camera calibration files had not been modified. However, assuming a temporal stable interior camera geometry is

unlikely amongst other due to the influence of temperature changes (Elias et al., 2020). In the future, refining the results will

require the ability to self-calibrate the camera (i.e., obtaining internal parameters) on-site during the analysis of each image

(e.g., as done and assessed in Elias et al., 2023).


Table 3 illustrates how cameras with shorter focal lengths exhibit greater re-projection errors. This is expected, as they tend to

have higher distortions, particularly at the edges of the image where the GCPs are located. Nevertheless, the results demonstrate

that the photogrammetric approach and calibration methods worked sufficiently. Notably, the case of NEU is significant, where

despite not having undergone field calibration, the re-projection values are comparable to those obtained at LAU. The

acquisition of GCP coordinates and camera calibration are crucial for achieving accurate exterior camera parameter estimation.

Errors at these steps affect the accuracy of the reprojection of the 3D model into the 2D space and the subsequent extraction

of the Z coordinates.

The creation of the 3D model using SfM algorithms proved to be effective, particularly during normal and high water levels.

In urban areas (e.g. GRO and NEU) the models are created using terrestrial cameras only, while at other study sites UAV

images are combined. In both cases, the riverbed has to be corrected for refraction influences. In GRO, a larger offset (i.e., -

2.5 cm) has to be applied to the wl_kiwa to minimise the discrepancy between the references zero heights due to the difficulties

in accurately modelling the riverbed due to significantly higher water levels on the day of image acquisition. For deep rivers,

where SfM cannot be applied because the riverbed is obscured, GNSS support for tracing cross-sections is necessary. In

general, a change in accuracy has been observed when the calculated water level intersects the area of the model that is

underwater, as the reconstruction quality of this zone is lower. Although this is not a critical issue, given that the system's

primary goal is to detect flooding, we suggest that the 3D model should be generated during periods of lower water levels to

improve the accuracies.

The intersection of the water body boundaries with the 3D model determines the calculated water level, making precision in

both elements crucial. The standard deviation of the extracted Z coordinates provides an indication of the quality of the

intersection, as we assume that the intersection should occur at a constant elevation (i.e., the same Z coordinate). Measurements

with high standard deviation suggest difficulty in estimating a stable Z value, which typically indicates either an erroneous

segmentation of the water pixels or that the water body-shore contact is not clearly visible in the image (e.g., vegetation

obscuring the contact, Figure 8d). However, the percentage of results filtered by standard deviation (Table 2) is not high,

demonstrating the robustness of the method.

The application of the Tukey filter based on official reference values has minimal impact on the total number of images used

and is only applied to remove erroneous values that may have passed through the standard deviation filter. Typically, these

errors are sporadic and also relate to poor segmentation (i.e., segmentation is performed in tiles that may produce horizontal

lines intersecting constant Z values), poor image visibility (e.g., presence of ice or animals on the camera), or a water level too

low leading to intersections with the 3D model in areas where reconstruction was not optimal.

Regarding the results obtained, we observe that the precision achieved in LAU is 1 cm in MAE during more than two years of

observation, indicating a very low average daily deviation from the official reference. This result confirms the roadmap for

future installations, as LAU is a location where the camera was calibrated on-site (i.e., low re-projection error), a moderately



large focal length was chosen (i.e., less distortion), and the water-shore contact zone is a concrete wall (i.e., segmentation without vegetation) situated at a distance of 15 m to the camera. In contrast, the ELB study site shows a MAE of 2.3 cm during 946 days of monitoring. This higher error is due to the lack of precise camera's internal parameters combined with a challenging

water-shore interface often obscured by vegetation, complicating segmentation. Figure 5 clearly shows abrupt changes in the determination of wl_kiwa, especially in summer, corresponding to vegetation maintenance in the area. This vegetation issue for image-based water level monitoring was also described in Eltner et al. (2018) and Peña-Haro et al. (2021). Especially for this location, we have found that there are often significant deviations from the official reference because just 100 m upstream there is a water mill, which regularly altered the water level in the measurement area in the dm-range within few minutes. The

official reference gauge averages the water level values every 15 minutes. Thus, every time the mill floodgate is opened or closed the averaged reference value does not represent the instant water level calculated by the camera gauge. Although we tried to minimise these effects using the daily averages, this regular discrepancy adds background noise. Finally, a MAE of 1.7 cm is obtained at GRO, where, although an on-site calibration was performed, difficulties arise due to the loss of GCPs, the width of the river and the distance to the camera (20 m), as well as the use of a wide-angle lens to capture the whole scene,

which adds complexity to the measurement workflow.

Another result demonstrating the need for maintenance (e.g. compute new 3D models, cut the grass, clean the study area, reinstall GCPs) and recalibration of the cameras is the evolution of precision over time. During the first 365 days (table 5), the precision obtained in LAU, ELB, and GRO is similar, i.e., around 1.5 cm in MAE, representing a benchmark of what can be

expected from a newly implemented low-cost camera system. However, over time it is not possible to maintain this level of accuracy. In the case of LAU, the MAE during the whole observation period is lower than during the first 365 days. This is due to the fact that the offset that minimises the differences with the reference is calculated for the entire time series, penalising, in this case, the first year of observation.

The results obtained represent an advance in accuracy, robustness and duration of observations compared to previous work also automatically estimating water levels in the same study sites. While Eltner et al. (2021) reported an error of -1.1 ± 3.1 cm for their best observation (using smoothing algorithms) at ELB, their best Spearman coefficient was 0.93, which is significantly lower than ours. In addition, the values presented by Eltner et al. (2021) showed significant seasonal variability, a problem we also noticed but which our AI approach resolved better. The method by Zamboni et al. (2025) requires less computational

power and avoids model training. However, they sacrifice precision, with mean absolute errors of 2.1 cm in LAU and 2.9 cm in ELB, and Spearman's correlations of 0.95 and 0.94, respectively. In addition, both papers prioritise methodological advances, without considering night-time observations, resilience to bad weather and time-series longevity. Compared to the work of Erfani et al. (2023), which covers only few hours of observation, our precision values remain in the same range as they obtain a minimum RMSE of 1.5 cm to 2.9 cm for the best observations at the most favourable side of the river.


In line with other published works that estimated water levels automatically from images and did not use measuring tapes, the results obtained in this study represent an advancement. For example, in Wang et al., 2024., Spearman coefficients range from 0.87 to a maximum of 0.94 for their best method, which covered an observation period of only two weeks. Vandaele et al., 2021, who analysed one year of observation, achieved Pearson correlations ranging from 0.94 to 0.96 for their best approach.

The Pearson correlations obtained in our work ranged from 0.96 at ELB to 0.99 at LAU, in image-by-image comparisons during a period that was twice as long as the one used by Vandale et al., 2021.

The results of this study are consistent with the German manual for river water level measurements, requiring water levels with accuracies below 2.0 cm  as an acceptable systematic error (Bund/Länderarbeitsgemeinschaft Wasser, 2018). The image-



by-image results obtained during the first year (Table 5) of observation comply with this requirement, including both the MAE and the RMSE. For the image-to-image results obtained over the entire time series (Table 4), both LAU and GRO consistently show results below 2.0 cm in terms of MAE. Only the ELB station revealed an error of 0.7 cm above the required limit. However, this error is not due to the methodology but rather to the specific installation site, which could be minimised by using a more responsive reference (i.e., that tolerate better rapid stream fluctuations), maintenance and proper and updated

camera calibration.

**5 Conclusion**

This study demonstrated the effectiveness of an AI-enhanced image-based camera gauge for long-term, near real-time river water level monitoring. Over a 2.5-year period, our approach was capable to accurately measure water levels with deviations below 2.5 cm, achieving Spearman correlation coefficients greater than 0.94 when compared to reference gauges. The usage

of neural networks for water segmentation and for GCP identification, combined with photogrammetry, allowed for the automatic processing of a large volume of images, even under adverse weather condition. The installation of IR lamps, combined with the surveillance cameras' ability to capture night time images, mitigated one of the major limitations of image-based methods, i.e., having measurements only during daylight, enabling 24/7 water level measurements. The results demonstrated increased inaccuracies over time, highlighting the need for proper maintenance of both the environment (e.g.,

vegetation) and the cameras (e.g., calibration). Future work aimed at minimizing these maintenance requirements, e.g., regenerating 3D models to adapt to terrain changes and automatically calibrating the cameras, will lead to further improvements in measurement quality and robustness.

**Video supplement:** The video supplement presents an animation of the water level measurements obtained at the Lauenstein gauging station from December 24 to 31, 2023, using the methodology described in this publication. It is accessible in the

Zenodo repository: https://doi.org/10.5281/zenodo.14875801 (Blanch et al., 2025b).

**Data availability**: The raw data and source code are not available for public access but can be provided upon request under reasonable conditions. The RIWA dataset used for the preliminary training of the dataset can be found in Blanch, X. et al., 2023.

**Competing interests**: The corresponding author has declared that none of the authors has any competing interests.

**Author contribution:** All authors contributed greatly to the work. XB: conceptualization, methodology, software, investigation, writing (original draft), figures, data acquisition; AE: conceptualization, methodology, writing (review and editing), supervision, funding acquisition; JG: conceptualization, supervision, writing (review), data acquisition, funding acquisition; RH: data acquisition, writing (review), supervision. All authors have read and agreed to the published version of the paper.

**Acknowledgements:** The authors would like to thank the Saxon state company for the environment and agriculture (Staatliche Betriebsgesellschaft für Umwelt und Landwirtschaft Sachsen), Germany for their support and cooperation in this study.

**Financial support**: The authors like to thank the ministry of education and research Germany for funding the KIWA project 'Artificial Intelligence for Flood Warning' (grant number 13N15542 and 13N15543) in the frame of the announcement 'Artificial Intelligence in Civil Security Research' as part of the Federal Government's 'Research for Civil Security'

programme.



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
