# Peer review of "AI Image-based method for a robust automatic real-time water level monitoring: A long-term application case"

_EGUsphere, 2025_

## Author Response (AR1)

**General comment to all reviewers**

The authors would like to thank the reviewers for the time and effort they dedicated to assessing our initial manuscript. The numerous and thoughtful comments covered a wide range of aspects, which resulted in the revision taking longer than we initially hoped. Nevertheless, the feedback provided enabled us to thoroughly re-examine our work and implement substantial changes to the manuscript—including improvements in writing, results, and figures—that we believe greatly benefit the publication. For this reason, the authors wish to express their sincere appreciation to the reviewers. We hope that the revised manuscript addresses all concerns and will be considered suitable for publication.

A point-by-point response to all reviewer comments is provided below:

- Reviewer comments are indicated in black text.

*- Author responses are indicated in blue text.*

*- New text added to the manuscript in response to your comments is indicated in orange.*

*- Extracts from the public response are indicated in green text.*

**Reviewer 1**

**General Evaluation:**

*Following the reviewer's recommendation, an Appendix has been added at the end of the manuscript, providing a complete list of acronyms. In addition, the Introduction has been revised in several parts to better define the research objectives and clarify the scientific framework of the proposed method. The final paragraph of the Introduction has also been rewritten to more clearly explain the overall context and rationale of the approach.*

*[line 138]: Rather than simply detecting flood events, our primary objective is to provide centimetre-precision water level measurements under both normal and high-flow regimes, operating robustly in real scenarios with real conditions. Although our approach requires greater technical complexity than simpler methods, this investment is justified because the georeferenced imagery establishes the foundation for future monitoring stages, enabling the derivation of surface velocity (amongst other by narrowing down the search area, i.e., the water mask) and discharge without additional in-stream instrumentation. By combining AI techniques for water segmentation and ground control point detection with established photogrammetric methods for image-to-geometry registration, the approach presented in this research will enable consistent and accurate water level monitoring over extended observation periods. Advancing beyond initial proof-of-concept testing (Eltner et al., 2018, 2021) this work presents a fully operational, contactless system showing 2.5 years of continuous data validated across multiple rivers and seasons, including adverse weather and nighttime conditions, and benchmarked against co-located official gauging stations.*

**Specific Comments:**

→ Page 4, Line 125: Move Figure 1 to the appropriate section "2.1 Study Area".

*Figure 1 has been relocated to Section 2.1 "Study Area" in the revised manuscript.*

→ Page 6, Line 150: What does "HWIMS" mean in Table 1?

*The term "HWIMS" has been removed and replaced with a clearer, more indicative name (wl_ref)*

→ Page 7, Line 186: In Figure 3, there are some unexplained acronyms (RIWA, KIWA?). Please include these in the proposed Appendix. Additionally, consider moving this figure to the beginning of the Methodology section, as it represents a flowchart of the entire procedure.

*An appendix including all acronyms has been added in the revised manuscript.*

→ Page 8, Line 192: In Figure 4, clarify the meaning of UperNet and ResNeXt50. Also, move this figure to the relevant section 2.4 "AI Segmentation".

*Figure 4 has been redesigned and moved to the corresponding Section 2.4 "AI Segmentation.".*

→ Page 9, Line 238: I noticed very high segmentation accuracy at night using IR. My question is: since the Sun also emits IR radiation, which is almost completely absorbed by water, would it be possible to simplify the procedure by using only IR intensities for water segmentation instead of training an AI model? If not, please highlight the advantages of your approach.

*A paragraph has been added discussing why IR imagery is not used under daylight conditions, as suggested:*

*[Line 441]: However, generalising the use of near-infrared imagery to daytime conditions instead of RGB—with the aim of simplifying segmentation—is not fully reliable. Daylight IR-only thresholding is unreliable because solar NIR and sunlight vary with illumination*

*geometry and surface reflections, which can reduce image sharpness and detail, whereas the nighttime approach operates under controlled illumination and yields more stable contrast .*

→ Page 9, Line 246: You mention that "the GCPs need to be measured in each image every time." Please explain more clearly the advantage of using GCPs instead of directly reading water levels from measuring tapes.

*A clearer explanation has been added in the Introduction regarding the advantages of using GCPs over direct scale-bar measurements:*

*[line 50:] For instance, one approach is to install scale bars in the observation area for an automatic measurement based on estimating the contact of the water with the scale bar (Kuo and Tai, 2022; Pan et al., 2018). These methods are highly efficient and provide good accuracy, but they require physical intervention in the river to install the scale bar. Moreover, scale bars and sensors installed under bridges or in direct contact with the water are vulnerable to mechanical damage, displacement by debris, and complete loss during flood events—precisely when accurate measurements are most critical. Progressive biofouling from algae and sediment accumulation further degrades visibility of graduations over time, necessitating frequent in-situ maintenance.*

*[Line 58]: In contrast, the contactless method proposed in this study eliminates the need for any in-stream installation, relying solely on a simple camera setup and ground control points (GCPs) positioned within the field of view. By interpreting the whole scene rather than a narrow scale window, this approach provides multi-point extraction across the segmented waterline, capturing spatially heterogeneous water level dynamics along the river cross-section rather than relying on single-point measurements that may not represent the true behaviour of the water surface. This inherent flexibility makes the system less sensitive to local occlusions or damage and more adaptable to site constraints and changing illumination. Furthermore, even when scale-bar methods are automated, reliable operation depends on optical character recognition (OCR) of small printed digits, which is highly susceptible to direct shadows, sun glare, water splashes, and oblique viewing angles that distort character shapes. By anchoring the entire scene in a georeferenced 3D frame, the method provides absolute water surface elevations and could potentially enable the analysis of complementary dynamic and morphological phenomena across the monitored area—capabilities that would remain unavailable in contact-based single-point instrumentation.*

→ Page 12, Line 332: MAE is a well-known metric; consider removing the formula or, if you decide to include it, format it properly rather than placing it within the text.

*The MAE formula has been removed from the revised manuscript.*

→ Page 12, Line 341: Move Table 3 closer to its first mention on page 11.

*Table 3 has been relocated closer to its first mention.*

→ Page 12, Line 347: In Table 4, explain all abbreviations used, or include them in the Appendix.

*All abbreviations in Table 4 are now explained in the Appendix.*

→ Page 13, Line 355: In Figure 5, clarify what the color scales indicate—are they related to segmentation, water levels, or something else?

*Figure 5 (now Figure 7) has been updated for improved clarity; its caption now explains the colour scales and includes a visible legend.*

→ Page 14, Line 379: Move Figure 6 to appear before the Discussion section, as it presents results.

*Figure 6 (now Figure 4) has been modified and moved to the Results section, as suggested.*

→ Page 15, Line 391: You state: "For future installations with different cameras and environments, we expect that only a small number of manually labelled images will be needed to adapt the model to the new site." I noticed that the selected river section features relatively uniform riverbed and bank characteristics. Is it accurate to say that a full retraining would not be needed even for sites with more diverse riverine environments? If so, please elaborate on this in the Discussion.

*An additional paragraph has been added in the Discussion to address this point:*

*[Line 414]: For future installations with different cameras and environments, only a small number of manually labelled images will be needed to adapt the model to the new site, because the bulk of the training data come from the RIWA dataset, which spans highly diverse aquatic domains (rivers, lakes, seas, urban environments, etc.). Adding a small percentage of site-specific images ensures maximum segmentation accuracy and, crucially, enables evaluation on the actual target domain, yielding a model tailored to the study area rather than a generalised classifier trained on unrelated scenes (e.g. coastal imagery). Furthermore, modern fine-tuning techniques that retrain only the final network layers can reduce deployment effort even further, potentially avoiding full retraining altogether. This transferability has been successfully demonstrated in practice by Krüger et al., (2024), where the workflow was retrained and applied in a very different environment in Oman, yet still delivered robust results.*

**Reviewer 2**

**Major comments:**

1.   The innovation of this study is not so clear. You propose a method for more accurate/robust water level measurement, right (Line 115)? But there is no comparison with the current measurement methods, and deep learning models from the literature? In the introduction section, I found there are some the-state-of-the-art techniques or models (e.g., SAM model).

*We have clarified the specific innovation and contribution of our research at the end of the Introduction. The Introduction has been revised to better highlight the novelty and the methodological framework of our approach. As mentioned during the public discussion, we do not consider it appropriate to perform direct comparisons between AI models, since this type of analysis has already been published in Wagner et al. (2023). Furthermore, we also already did performance evaluations with SAM (Zamboni et al., 2025) and therefore do see not added value repeating this. We have also added a note explaining why our comparison focuses on official measurements rather than alternative measuring devices.*

*[line 137]: Rather than simply detecting flood events, our primary objective is to provide centimetre-precision water level measurements under both normal and high-flow regimes, operating robustly in real scenarios with real conditions. Although our approach requires greater technical complexity than simpler methods, this investment is justified because the georeferenced imagery establishes the foundation for future monitoring stages, enabling the derivation of surface velocity (amongst other by narrowing down the search area, i.e., the water mask) and discharge without additional in-stream instrumentation. By combining AI techniques for water segmentation and ground control point detection with established photogrammetric methods for image-to-geometry registration, the approach presented in this research will enable consistent and accurate water level monitoring over extended observation periods. Advancing beyond initial proof-of-concept testing (Eltner et al., 2018, 2021) this work presents a fully operational, contactless system showing 2.5 years of continuous data validated across multiple rivers and seasons, including adverse weather and nighttime conditions, and benchmarked against co-located official gauging stations.*

*[line 165]: At the three gauged sites, water level measurements derived from the camera system (wl_opt) are validated against official gauge records (wl_ref). These reference measurements are obtained by float-operated and bubble gauges (redundant measurement systems) and provide the most appropriate benchmark for operational validation because they represent independent, quality-controlled records reflecting accepted hydrometric practice (ISO 18365, 2013) and are used directly by flood services for decision-making. This comparison isolates the performance of the camera-based workflow from site-specific factors (e.g., camera placement, local calibration) and ensures the evaluation aligns with operational monitoring standards.*

2.   There is a lack of report and analysis of accuracy on different scenarios (different weather, transparent water, vegetation cover....). This can be done by dividing test images into several subsets with different scenarios. Then, test the model on them to evaluate the generalization ability across different scenarios. Without this analysis, it is hard to say the method is robust.

*As discussed during the public response, the authors do not consider additional modifications necessary in this regard. The manuscript already demonstrates the robustness of the proposed method through the presented results, which span a two-year monitoring period and include images acquired under a wide range of environmental conditions. Furthermore, the original manuscript already specifies the definition of robustness adopted in this study:*
*[line 100]: The work presented here addresses these challenges by meeting the robustness criteria defined by Peña-Haro et al. (2021), as it achieves key properties such as continuous image capture throughout the whole day, applicability across different rivers, and the capacity for near real-time data transmission and processing."*

3.   While so many images (200k) are used to evaluate models, the generalization ability of models to an unseen location is not explored or discussed. That is important for real operation, e.g., applying this model to a different river or a new country with limited labeled data and real water level measurement. That can be done by training models on one location and test the generalization to other locations. Otherwise, it is hard to say the method is robust.

*The original manuscript already discusses the possibility of generalizing the proposed workflow to new study areas. We have never indicated that the method can be directly applied to new sites without retraining. As we mentioned earlier, linking the robustness criterion to the ability to apply the method at unseen locations without any retraining seems arbitrary to us. A new paragraph has been added to clearly clarify the limits of the method in this respect.*

*[Line 414]: For future installations with different cameras and environments, only a small number of manually labelled images will be needed to adapt the model to the new site, because the bulk of the training data come from the RIWA dataset, which spans highly diverse aquatic domains (rivers, lakes, seas, urban environments, etc.). Adding a small percentage of site-specific images ensures maximum segmentation accuracy and, crucially, enables evaluation on the actual target domain, yielding a model tailored to the study area rather than a generalised classifier trained on unrelated scenes (e.g. coastal imagery). Furthermore, modern fine-tuning techniques that retrain only the final network layers can reduce deployment effort even further, potentially avoiding full retraining altogether. This transferability has been successfully demonstrated in practice by Krüger et al., (2024), where the workflow was retrained and applied in a very different environment in Oman, yet still delivered robust results.*

4.   I found it is a quite complex method for automatically estimating river water levels. For example, one needs to build a 3D model using images from drones (Section 2.2) and measure the GCPs in each image every time (Line 246). That needs lots of expert knowledge,

human labors, and specific equipment. What is the advantage of the method. It is hard to say it is "low-cost" in abstract and conclusion. Same with my first major comment, you need to estimate the cost of different methods in the literature. How about using a scale bar combined with the computer vision system? While it requires intervention in the river, it is low-cost and straightforward to show the water level without post-processing.

*As discussed in the public response, the goal of this paper is not to review or compare existing methods, but to present a new camera-based approach validated against official gauge measurements. A new paragraph has been added to better describe the main advantages of the proposed method. In addition, the last two paragraphs of the Introduction have been revised to further clarify the research goals and to justify the methodological "complexity." As previously mentioned in the public discussion, we do not consider the level of complexity to be a valid criterion for evaluating the relevance or soundness of the proposed approach.*

*[line 58]: In contrast, the contactless method proposed in this study eliminates the need for any in-stream installation, relying solely on a simple camera setup and ground control points (GCPs) positioned within the field of view. By interpreting the whole scene rather than a narrow scale window, this approach provides multi-point extraction across the segmented waterline, capturing spatially heterogeneous water level dynamics along the river cross-section rather than relying on single-point measurements that may not represent the true behaviour of the water surface. This inherent flexibility makes the system less sensitive to local occlusions or damage and more adaptable to site constraints and changing illumination. Furthermore, even when scale-bar methods are automated, reliable operation depends on optical character recognition (OCR) of small printed digits, which is highly susceptible to direct shadows, sun glare, water splashes, and oblique viewing angles that distort character shapes. By anchoring the entire scene in a georeferenced 3D frame, the method provides absolute water surface elevations and could potentially enable the analysis of complementary dynamic and morphological phenomena across the monitored area—capabilities that would remain unavailable in contact-based single-point instrumentation.*

*[line 125]: Despite all these recent advances in AI and image-based water level monitoring, a critical gap remains: no existing image-based contactless method has demonstrated operational reliability over extended periods (>1 year) across multiple sites under full 24/7 conditions, including adverse weather and nighttime. This research addresses this gap through the KIWA project (Künstliche Intelligenz für die Hochwasserwarnung - Artificial Intelligence for Flood Warning) (Grundmann et al., 2024), which aims to develop AI-driven tools for comprehensive flood monitoring and early warning systems. Within this framework, our research addresses the foundational requirement: accurately quantifying river water levels in a fully automated, contactless manner.*

*Rather than simply detecting flood events, our primary objective is to provide centimetre-precision water level measurements under both normal and high-flow regimes, operating robustly in real scenarios with real conditions. Although our approach requires greater technical complexity than simpler methods, this investment is justified because the georeferenced imagery establishes the foundation for future monitoring stages, enabling the derivation of surface velocity and discharge without additional in-stream instrumentation. By combining AI techniques for water segmentation and ground control point detection with established photogrammetric methods for image-to-geometry registration, the approach presented in this research will enable consistent and accurate water level monitoring over extended observation periods. Advancing beyond initial proof-of-concept testing (Eltner et al., 2018, 2021) this work presents a fully operational, contactless system showing 2.5 years of continuous data validated across multiple rivers and seasons, including adverse weather and nighttime conditions, and benchmarked against co-located official gauging stations.*

**Minor comments:**

1. Lacking an overall introduction of the methodology before Section 2.2. Please present the procedure of the overall methodology, and the link with each method or Section (i.e., 2.2-2.5). It can be linked to the Fig. 2. By the way, you have two "Section 2.5".

   *We have added a brief introductory paragraph at the beginning of the Methodology section to provide an overview of the workflow and clarify its connection with the subsequent subsections (2.2–2.5), as suggested by the reviewer. In addition, the duplicated numbering of Section 2.5 has been corrected..*

   *[line 144]: Achieving robust, 24/7 water level monitoring without in-stream installations requires integrating multiple disciplines. Field surveys and geodesy establish the metric reference frame, photogrammetry enables contactless measurements through image-to-geometry registration, artificial intelligence automates feature extraction under varying conditions, and software engineering orchestrates real-time processing. The workflow comprises three interdependent stages: establishing a georeferenced 3D site model with ground control points (GCPs) visible in the camera's field of view; automated image processing using AI algorithms for river segmentation and GCP detection; and photogrammetric reprojection of segmented water onto the 3D model to derive metric water levels. This integration enables near-real-time operation without manual intervention, as detailed in the following subsections. Figure 1 illustrates the methodological workflow, while Figure B1 in Appendix B presents a detailed algorithm flowchart from image acquisition to the final water level measurement (wl_kiwa) and comparison against the reference gauge value (wl_ref); all acronyms, especially those related to the project, are defined in Appendix A.*

2. Line 205, lack the names of 32 CNN architectures. Possibly report them in Appendix.

   *We decided not to include the detailed list of the 32 CNN architectures, as they belong to a different publication by other authors. To avoid potential confusion, the reference to the 32 CNNs (including SAM, as mentioned by the reviewer) has been removed from line 205, since their evaluation was not part of the present study and may have led to a misunderstanding.*

*3.*   Line 225, what data augmentation technique do you use. Provide details.

*The paragraph has been rephrased to improve clarity. Authors consider it unnecessary to provide additional details:*

*[line 240]: Both datasets are augmented using the Albumentation library (Buslaev et al., 2020), to increase training robustness and cross-site transferability. Augmentation is particularly critical for fixed-camera systems to prevent overfitting due to spatial correlations inherent in images from static viewpoints. Geometric transformations (rotation, scaling, flipping) were applied to break spatial dependencies, while pixel-level modifications (brightness, contrast, hue adjustments) were constrained to realistic ranges to preserve natural image appearance. Data augmentation was applied after generating all 512×512 patches from each original image in the training dataset, with each original patch producing four augmented versions.*

4.   Line 225, what is the size of images used for training and testing? Raw images or image patch (512*512) or both? It is unclear. Why not perform data augmentation on raw images?

*We believe that the original text is sufficiently clear. The expression "512×512 patches from each original image" refers to patches (i.e., portions of the full images), not to entire images. Applying augmentation to the raw images rather than to the patches provides less variability, as augmenting 100 full images and then tiling them is not equivalent to augmenting each individual patch derived from those images.*

5.   I suggest labelling subfigure (a), (b)…on the Fig.2, and present the specific the subfigure in the text, rather than Fig. 2 for more readable (e.g., in Line 245).

*Done. Subfigures have been labelled accordingly in Figure 2 (now Figure 1) and referenced in the text for improved readability.*

6.   Line 225, you did not state the model specifically trained on KIWA images above. Is it the model trained on night images along? The readers miss the information of this model. How you train this model? How many training images? What model architecture? They are not mentioned before.

*We are not entirely sure what the reviewer is referring to, since line 225 already describes the data augmentation procedure (see Comment 4). If, instead, the remark concerns the training on night-time images, the relevant paragraph has been rephrased to avoid any distinction between day and night imagery. In addition, Figure 3 has been updated to clearly specify the networks used. We have also modified:*

*[line 239] Due to the absence of publicly available IR river datasets, this training relied on 146 manually annotated KIWA IR images capturing varied weather and water level conditions (Figure 3b). .*

*and we have now added the following lines to make the description symmetric with the daytime case:*

*[line 252] The nighttime model was evaluated exclusively on KIWA images due to the lack of external IR datasets. The best-performing daytime model achieved 98.9% pixel-wise accuracy, while the nighttime model reached 99.1% accuracy, where accuracy represents the proportion of pixels correctly classified as water or non-water.*

7.   Lack an overall description of Fig. 3 in the text.

*Thanks, Figure 3 (now Figure B1) has been moved to Appendix (following the suggested structure by coopernicus). And a better description has ben included in the manuscript:*

*[line 156]: This integration enables near-real-time operation without manual intervention, as detailed in the following subsections. Figure 1 illustrates the overall methodological workflow, whereas Figure B1 in Appendix B provides a detailed algorithm flowchart covering all steps from image acquisition to the final water-level estimation obtained with the optical system (wl_opt) and its comparison with the reference gauge value (wl_ref). All acronyms, including those specific to the project, are defined in Appendix A.*

8.   Line 30, before stating the advantages of images-based techniques, I suggest stating the state-of-the-art of the non-image-based techniques and their applications and disadvantages

*The authors have decided not to implement this minor suggestion. As explained in the public response, the goal of this work is to advance image-based methods, and we believe that the opening of the paper ("The use of image-based systems has transformed the field of geosciences…") already conveys this clearly. This manuscript is not intended to discuss or compare the advantages and disadvantages of non–image-based water-level estimation techniques. Furthermore, following the recommendation of another reviewer, we have slightly shortened the Introduction section.*

9.   Line 35, provide several examples of these applications, not just state "water management", which is a very big scope

*We have rephrased the paragraph and added three examples:*

*[line 35] These advantages make image based systems an optimal tool for the management and study of water resources—encompassing, for example, turbidity assessment (Miglino et al., 2025; Zhou et al., 2024), water-discharge estimation (Eltner et al., 2020), and flood detection (Fernandes et al., 2022), —thereby increasing the capacity to respond to extreme events and facilitating evidence based decision making in water management.*

10. Line 45, what is hydrological monitoring? provide some examples
11.

*This clarification is no longer necessary. The paragraph has been removed, as the authors consider that it did not provide a specific framework for introducing the conducted work. Following the recommendations of other reviewers, the Introduction has also been streamlined.*

12. Line 55, what is traditional methods? Please clarify

*It was a mistake, traditional means "basic optical-based methods"*

*[line 47] Basic optical-based methods often struggle with long-term continuous operation or automation, particularly during adverse weather conditions or at night.*

13. Line 55, cameras also need continuous operation, such as changing the battery or SD memory card.

*While battery or SD-card replacement can be a limitation in stand-alone setups, continuous camera-based deployments for environmental monitoring (as the equipment deployed in our research) typically employ battery/solar power and cellular (e.g., 4G/LTE) connectivity to enable unattended, long-term operation.*

*Moreover, the manuscript already specifies that our cameras are surveillance cameras that are permanently powered and network-connected—and that images are transmitted directly to the servers.*

14. Line 55, you provided some citaions. Do they use camera and AI? Please clarify what is the automatic measurement here?

*I am not entirely sure of the intent of this comment, because reviewer 2's comments 11, 12, 13, and 14 all refer to line 55. The paragraph containing line 55 does not discuss artificial intelligence; it only addresses methods for obtaining water level using image-based methods to fit and justify our research. And it also provides two examples (Kuo and Tai, 2022; Pan et al., 2018) in which a scale bar installed in the river is captured by an optical method.*

*As mentioned in comment Reviewer #2, comment 10. We have removed the paragraph regarding AI for hidrological monitoring.*

15. Line 55, How much is the affect of this intervention (i.e., scale bar)? It could be a small stick installed near the bank, with little affect to the river,right? Please clarify it

*If the comment refers to other works, our goal as researchers is not to quantify the impact of third-party interventions. We identify the limitations—discussed in detail in the manuscript—and propose a method intended to overcome them. It would not be meaningful to analyze a specific intervention implemented by other researchers, especially considering that thousands of scale bars are installed worldwide. As mentioned in Comment 4 of the major comments, we have added a paragraph to clarify the advantages of our system compared to other approaches. For instance, a small stick or scale bar could easily be covered by branches, vegetation, or other debris during a flood event.*

16. Line 115, you said "their accuracy limitations make them unsuitable for reliable water level monitoring". what is the detailed accuracy of the previous studies. Show metrics. And what level of accuracy is suitable?

*Once again, this point has already been addressed in detail by Zamboni et al. (2025) and Wagner et al. (2023) that work extensively with AI networks to segment water. In our research we do not use any "generic AI approach.". We consider that it is not reasonable to ask the authors of this manuscript to reproduce "detailed" work and technical parameters reported by other studies that are correctly well cited.*

*We are not to include that in the manuscript but this is an extraction from Zamboni et al. (2025) conclusions:
"Qualitative analysis of images from the camera gauge stations showed that STCN generally achieved a better result than SAM Dino and SAM Six Points, especially for the river borders".*

**Reviewer 3**

**MAIN ISSUES**

1. Scientific Article vs. Scientific Report

While the technical work is solid, the manuscript often reads more like a project report than a scientific article. It lacks a focused research question, has an overly descriptive tone, and does not critically reflect on methodological generalizability—especially beyond the AI component.

*We thank the reviewer for this valuable comment. In response, the entire Methodology section has been thoroughly revised to adopt a more concise and analytical tone, in line with the standards of a scientific article rather than a project report. The revised manuscript now presents a focused research question at the end of the Introduction, and the methodology emphasizes generalizability and critical assessment throughout. In addition, new figures and quantitative analyses have been introduced to reinforce the manuscript's scientific contribution. We trust these substantial improvements address the reviewer's concerns regarding clarity, style, and scientific focus.*

*[line 149 to line 300]*

2. Transferability is Overstated

The authors claim high model transferability, but the full system depends on site-specific 3D modeling, GCP placement, camera calibration, and environmental maintenance (e.g., vegetation control). The workflow's performance degrades over time without intervention. The authors should distinguish between model-level transferability (e.g., CNN reuse) and system-level replicability, which is far more constrained.

*The authors do not see a direct correlation between the potential performance degradation over time without maintenance and the concept of model transferability (which we do not claim as full or universal). We believe that the original manuscript already provides a clear and honest discussion of the method's limitations. In line with our public response and previous reviewer comments, the corresponding paragraph has been revised to clarify the statement regarding "future installations."*

*[Line 414]: For future installations with different cameras and environments, only a small number of manually labelled images will be needed to adapt the model to the new site, because the bulk of the training data come from the RIWA dataset, which spans highly diverse aquatic domains (rivers, lakes, seas, urban environments, etc.). Adding a small percentage of site-specific images ensures maximum segmentation accuracy and, crucially, enables evaluation on the actual target domain, yielding a model tailored to the study area rather than a generalised classifier trained on unrelated scenes (e.g. coastal imagery). Furthermore, modern fine-tuning techniques that retrain only the final network layers can reduce deployment effort even further, potentially avoiding full retraining altogether. This transferability has been successfully demonstrated in practice by Krüger et al., (2024), where the workflow was retrained and applied in a very different environment in Oman, yet still delivered robust results.*

3. Ambiguity of Goal

Around line 455, the authors state that the system's primary goal is flood detection. However, the efforts seem instead directed on achieving very high accuracy (i.e., ≤2cm in line with German standards). I gather these are distinct goals with different technical demands. If the intended application is flood detection, simpler models could likely achieve acceptable performance without the need for full 3D reconstruction.

*As stated in the public response, we have removed the specific reference to flood warning. As mentioned in previous comments, several parts of the Introduction have been edited, and a final paragraph has been added to clarify the overall goal of the project and our contribution within it.*

*[line 130]: Despite all these recent advances in AI and image-based water level monitoring, a critical gap remains: no existing image-based contactless method has demonstrated operational reliability over extended periods (>1 year) across multiple sites under full 24/7 conditions, including adverse weather and nighttime. This research addresses this gap through the KIWA project (Künstliche Intelligenz für die Hochwasserwarnung - Artificial Intelligence for Flood Warning) (Grundmann et al., 2024), which aims to develop AI-driven tools for comprehensive flood monitoring and early warning systems. Within this framework, our research addresses the foundational requirement: accurately quantifying river water levels in a fully automated, contactless manner as a basis for also measuring surface velocity and discharge optically — all derived from a single sensor, the camera.*

*Rather than simply detecting flood events, our primary objective is to provide centimetre-precision water level measurements under both normal and high-flow regimes, operating robustly in real scenarios with real conditions. Although our approach requires greater technical complexity than simpler methods, this investment is justified because the georeferenced imagery establishes the foundation for future monitoring stages, enabling the derivation of surface velocity (amongst other by narrowing down the search area, i.e., the water mask) and discharge without additional in-stream instrumentation. By combining AI techniques for water segmentation and ground control point detection with established photogrammetric methods for image-to-geometry registration, the approach presented in this research will enable consistent and accurate water level monitoring over extended observation periods. Advancing beyond initial proof-of-concept testing (Eltner et al., 2018, 2021) this work presents a fully operational, contactless system showing 2.5 years of continuous data validated across multiple rivers and seasons, including adverse weather and nighttime conditions, and benchmarked against co-located official gauging stations.*

4. Comparisons

The comparisons made to prior work are not entirely fair or consistent. Don't the datasets differ in river conditions, locations, and environmental conditions? Moreover, some of the simpler methods perform comparably in terms of key metrics. If the actual goal is flood detection, why shouldn't we use these methods instead? Furthermore, given the costly efforts in maintenance, how does this method compare against non-computer vision based approaches? What are the real costs/benefits of deploying and maintaining this system vs alternatives?

*As answered in the public document, we not agree with this statement:*

> *[...We respectfully disagree with the suggestion that our comparisons are unfair. Based on the reviewer's comment, it might seem that our manuscript criticizes or diminishes other methods, which was never our intention. We will review the manuscript to ensure this is not the case and will clarify in the introduction that our aim is to present a new method that addresses some of the limitations of previous approaches—particularly by offering a fully contactless solution.*
>
> *We do not believe it is appropriate to compare our method directly with approaches focused solely on flood detection, as our development clearly extends beyond that specific application. As noted in our response to Reviewer #2, a comprehensive comparison with all non-image-based methods would be more suitable for a review article than for a research paper focused on a specific methodological contribution.*
>
> *We would see the need for exhaustive comparisons if our work were limited to laboratory development under controlled conditions. However, we believe that benchmarking our system against official gauge stations, operating in the field for over two years, offers the most meaningful and rigorous evaluation...]*

5. Evaluation Metrics and Goals

The paper reports absolute errors and Spearman correlations but lacks relative error analysis or discussion of operational thresholds. Again, if flood detection is the goal, is it really necessary to optimize for every centimeter? Perhaps the authors should provide more context about what performance is "good enough" for different use cases?

*We appreciate the reviewer's thoughtful comment. We appreciate the reviewer's thoughtful comment. In the revised manuscript, we have included several additional performance metrics beyond Spearman correlation, such as mean absolute error (MAE), symmetric mean absolute percentage error (sMAPE), and Nash-Sutcliffe efficiency (NSE), which are now illustrated and supported by new plots in Figures 5 and 6. This provides a more comprehensive characterization of the model's operational error, both in absolute and relative terms, for all study sites. The optimization at the centimeter level may or may not be necessary depending on the intended application, but it should not constrain the methodological development or its scientific investigation. The paper already discusses in detail the factors that can lead to higher or lower absolute errors. We also consider that defining operational thresholds falls outside the scope of this research, as such thresholds depend on the specific objectives of the end users and the particular design of each deployment.*

**MINOR COMMENTS**

> Several figures (e.g., Figures 5, 6, and 8) are difficult to interpret due to visual clutter or low resolution. Figure 5 tries to convey too many sites at once; Figure 6 requires zooming to read. Consider simplifying visuals, focusing on a single site per panel, or moving supporting visuals to an appendix.

*We thank the reviewer for this suggestion. Figures 5, 6, and 8 have been modified to emphasize results from a single study area, thereby simplifying the visuals. The timeline figure combining results for three sites has been retained but its rendering has been improved for greater clarity. Following Copernicus editorial guidelines, similar figures corresponding to other sites have been moved to the Appendix. All new figures employ the same scheme and now present examples from only one study area in the main text, with corresponding figures for other locations included in the Appendix for reference.*

> Terms like "KIWA variables" are not broadly meaningful outside the context of this specific project. Using more general terminology would improve clarity and support reproducibility in other settings.

*We understand the reviewer's concern. In this regard, we have created Appendix A (in response to Reviewer 1's comment) and have removed all acronyms linked to KIWA. For instance, the previous "wl_kiwa" variable is now replaced by "wl_opt" (from optical-based measurement).*

> The paper states that code and data are only available upon request. Given that the authors build on standard architectures (UPerNet, R-CNN), and claim reproducibility, it is unclear why the full codebase is not released. Sharing trained models and AI pipelines would improve transparency. Is this link to commercial efforts?

*Thank you for raising this point. In the revised version, all source code is now available in a Zenodo repository, and the trained AI models used for this study have been deposited as well. These resources are openly accessible to support full transparency and reproducibility. As stated in our public response, we wish to clarify that there is no commercial involvement connected to this research.*

> Line 105 – The hyperlink for Wagner incorrectly includes the word "IN" as part of the link. This should be corrected for proper formatting and readability.

*Done. The hyperlink for Wagner has been corrected.*

> Line 121 – The discussion of Ground Control Point (GCP) detection around this line appears without sufficient context.

*Thanks for this comment. It's true, we removed the GCPs reference here because is confused and is not presented before!*

> The introduction is long and could be trimmed. The contributions should be clearly stated toward the end of the introduction, rather than dispersed throughout. Also, I do not see sufficient framing with respect to other methods than Computer Vision.

*We find this request somewhat conflicting with other reviewers' guidance to expand the Introduction and to elaborate further on the state of the art of alternative approaches. Our view is that the Introduction should summarize the state of the art pertinent to the methods employed in our manuscript and provide only the background necessary to contextualize our contribution. But accordingly with this reviewer, we have removed a paragraph that was considered to provide unnecessary contextual information.*

> The section currently labeled Methodology also covers the study area and data. Consider renaming it to Methods and Materials to better reflect the content. Also, this section is the hardest to read and also gives the feeling we are reading a technical report, not a scientific paper.

*We have retitled the section from "Methodology" to "Materials and Methods." And, as suggested by previous reviewers, we added a brief introductory paragraph to improve readability and to avoid a technical-report style.*